# Bäcklund Transformations for Liouville Equations with Exponential Nonlinearity

Tatyana V. Redkina [1,2], Robert G. Zakinyan [2,3,*], Arthur R. Zakinyan [2,3] and Olga V. Novikova [2,4]

1 Faculty of Mathematics and Computer Sciences Named after Professor N. I. Chervyakov, North-Caucasus Federal University, 1 Pushkin Street, 355017 Stavropol, Russia; TVR59@mail.ru
2 North-Caucasus Center for Mathematical Research, 1 Pushkin Street, 355017 Stavropol, Russia; zakinyan.a.r@mail.ru (A.R.Z.); oly-novikova@yandex.ru (O.V.N.)
3 Physical-Technical Faculty, North-Caucasus Federal University, 1 Pushkin Street, 355017 Stavropol, Russia
4 Institute for Digital Development, North-Caucasus Federal University, 2 Kulakov Avenue, 355017 Stavropol, Russia
* Correspondence: zakinyan@mail.ru; Tel.: +7-918-7788-675

**Abstract:** This work aims to obtain new transformations and auto-Bäcklund transformations for generalized Liouville equations with exponential nonlinearity having a factor depending on the first derivatives. This paper discusses the construction of Bäcklund transformations for nonlinear partial second-order derivatives of the soliton type with logarithmic nonlinearity and hyperbolic linear parts. The construction of transformations is based on the method proposed by Clairin for second-order equations of the Monge–Ampere type. For the equations studied in the article, using the Bäcklund transformations, new equations are found, which make it possible to find solutions to the original nonlinear equations and reveal the internal connections between various integrable equations.

**Keywords:** nonlinear equations in partial derivatives; hyperbolic equations; Bäcklund transformations; Clairin's method; differential relationships; the Liouville equation

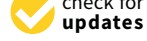



## 1. Introduction

The study of Bäcklund transformations is one of the current topics in the theory of partial differential equations. Such transformations are used to find solutions to nonlinear differential equations. Due to the complexity of various nonlinear equations, there is no single method for solving them. For integrable systems, effective methods have been developed, such as the inverse scattering method [1,2], the Hirota method [3–5], the Painleve method [6,7], Bäcklund transformations [8–11] and the mapping and deformation method [2].

Bäcklund transformations are an example of differential geometric structures generated by differential equations. They make it possible to obtain not only pairs of equations but also a solution to one of them if the solution to the other is known. These transformations play an important role in integrable systems since they reveal internal connections between various properties, such as the definition of symmetries [12,13] and the presence of a Hamiltonian structure [14–16]. More recently, many studies have been carried out in this area [11,17–19].

This article is a presentation of new results on transformations and auto-Bäcklund transformations for equations of the Klein–Gordon type, using the method of constructing transformations for the Liouville equation. The paper considers special cases of equations with exponential–power nonlinearity having a factor depending on the first derivatives. The construction of transformations is based on Clairin's method [20].

## 2. Methods

We consider the following nonlinear equation of the hyperbolic form:

$$v_{\xi\eta} = f(v, v_\xi, v_\eta). \tag{1}$$

The method developed by Clairin to construct Bäcklund transformations of a general form is applicable when the functions $z$ and $v$ satisfy different partial differential equations. The technique of constructing Bäcklund transformations is general to any hyperbolic equation and completely repeats the construction for the Liouville equation.

Differential equations of the second order of the form

$$f_1(\xi, \eta, z, z_\xi, z_\eta)\, z_{\xi\xi} + f_2(\xi, \eta, z, z_\xi, z_\eta)\, z_{\xi\eta} + f_3(\xi, \eta, z, z_\xi, z_\eta)\, z_{\eta\eta} + f_4(\xi, \eta, z, z_\xi, z_\eta) = 0.$$

are called Monge–Ampere equations [21]. The Bäcklund transformation linking two such second-order equations for the $v$ and $z$ functions is given by a pair of first-order differential equations:

$$\frac{\partial z}{\partial \xi} = F_1\left(z, v, \frac{\partial v}{\partial \xi}, \frac{\partial v}{\partial \eta}\right). \tag{2}$$

$$\frac{\partial z}{\partial \eta} = F_2\left(z, v, \frac{\partial v}{\partial \xi}, \frac{\partial v}{\partial \eta}\right). \tag{3}$$

To define an explicit transformation type, it is necessary to find the functions $F_1$ and $F_2$. The integrability condition (the equality of the mixed second derivatives) requires that the functions (2), (3) satisfy the relation

$$\frac{\partial^2 z}{\partial \eta \partial \xi} - \frac{\partial^2 z}{\partial \xi \partial \eta} = 0.$$

Each of the variables $z$, $z_\xi$, $z_\eta$ and, respectively, $v$, $v_\xi$, $v_\eta$, depends on $\xi$ and $\eta$. Given the equality (2), we obtain

$$\frac{\partial^2 z}{\partial \eta \partial \xi} = \frac{\partial F_1}{\partial \eta} = \frac{\partial F_1}{\partial z} z_\eta + \frac{\partial F_1}{\partial v} v_\eta + \frac{\partial F_1}{\partial v_\xi} v_{\xi\eta} + \frac{\partial F_1}{\partial v_\eta} v_{\eta\eta} \tag{4}$$

$$\frac{\partial^2 z}{\partial \xi \partial \eta} = \frac{\partial F_2}{\partial \xi} = \frac{\partial F_2}{\partial z} z_\xi + \frac{\partial F_2}{\partial v} v_\xi + \frac{\partial F_2}{\partial v_\xi} v_{\xi\xi} + \frac{\partial F_2}{\partial v_\eta} v_{\eta\xi} \tag{5}$$

Using Formulas (2), (3) to exclude $z_\xi$ and $z_\eta$, finally, we obtain the condition of compatibility as

$$\left(-\frac{\partial F_2}{\partial v_\xi}\right) v_{\xi\xi} + \left(\frac{\partial F_1}{\partial v_\xi} - \frac{\partial F_2}{\partial v_\eta}\right) v_{\xi\eta} + \frac{\partial F_1}{\partial v_\eta} v_{\eta\eta} - \frac{\partial F_2}{\partial v} v_\xi + \frac{\partial F_1}{\partial v} v_\eta + F_2 \frac{\partial F_1}{\partial z} - F_1 \frac{\partial F_2}{\partial z} = 0 \tag{6}$$

We consider the function $z$ as a solution to some simple equation, the form of which is defined below. Then, while at least one of the coefficients,

$$\frac{\partial F_1}{\partial v_\eta}, \ \frac{\partial F_2}{\partial v_\xi} \ \text{or} \ \left(\frac{\partial F_1}{\partial v_\xi} - \frac{\partial F_2}{\partial v_\eta}\right),$$

is not zero, Equation (6) is a partial differential equation for the function $v$.

Since Equation (1) contains $v_{\xi\eta}$, but not $v_{\xi\xi}$ or $v_{\eta\eta}$, from the condition of compatibility (6), we expect that

$$\frac{\partial F_2}{\partial v_\xi} = 0, \ \frac{\partial F_1}{\partial v_\eta} = 0, \ \frac{\partial F_1}{\partial v_\xi} - \frac{\partial F_2}{\partial v_\eta} \neq 0$$

Then, we must assume

$$\frac{\partial z}{\partial \xi} = F_1\left(z, v, \frac{\partial v}{\partial \xi}\right), \tag{7}$$

$$\frac{\partial z}{\partial \eta} = F_2\left(z, v, \frac{\partial v}{\partial \eta}\right). \tag{8}$$

Therefore, Equation (6) takes the form

$$\left(\frac{\partial F_1}{\partial v_\xi} - \frac{\partial F_2}{\partial v_\eta}\right) v_{\xi\eta} - \frac{\partial F_2}{\partial v} v_\xi + \frac{\partial F_1}{\partial v} v_\eta + F_2 \frac{\partial F_1}{\partial z} - F_1 \frac{\partial F_2}{\partial z} = 0$$

The $\eta$-derivative of (7) is

$$\frac{\partial^2 z}{\partial \eta \partial \xi} = \frac{\partial F_1}{\partial z} z_\eta + \frac{\partial F_1}{\partial \tilde{z}} v_\eta + \frac{\partial F_1}{\partial v_\xi} v_{\xi\eta}. \tag{9}$$

Further reasoning depends on the type of equation under consideration. Let us consider the following equations:

$$v_{\eta\xi} = (a + bv)e^v v_\xi - v_\xi v_\eta, \tag{10}$$

$$v_{\eta\xi} = \frac{\alpha_{32}\alpha_{21}}{8\alpha_{11}} e^v (1 + 2v) v_\xi - v_\xi v_\eta, \tag{11}$$

$$v_{\eta\xi} = \frac{\alpha_{21}}{8\alpha_{11}^2} e^v (v_\eta - v_\xi), \tag{12}$$

$$v_{\eta\xi} = e^v v_\eta - e^{-v} v_\xi. \tag{13}$$

These equations have a hyperbolic linear form on the left side and a nonlinear right side depending on the function and the first derivatives to variables $\eta$ and $\xi$, wherein the derivatives $v_\eta, v_\xi$ are included in equations only in the first degree, so the general form of these equations is rewritten as

$$v_{\eta\xi} = \Omega(v, v_\xi^1, v_\eta^1),$$

Here, a one in the exponent indicates that these variables are included in this equality only to the first degree.

We assume that the Bäcklund transformations make it possible to move to the simplest hyperbolic equation $z_{\xi\eta} = 0$.

Using Equations (8)–(10), we obtain

$$z_{\xi\eta} = \frac{\partial F_1}{\partial z} F_2 + \frac{\partial F_1}{\partial v} v_\eta + \frac{\partial F_1}{\partial v_\xi} \Omega(v, v_\xi^1, v_\eta^1) = 0. \tag{14}$$

Take from (14) the derivative to $v_\eta$. Then, $\frac{\partial \Omega(v, v_\xi^1, v_\eta^1)}{\partial v_\eta}$ does not depend on $v_\eta$, since $v_\eta$ comes into equality only in the first degree

$$\frac{\partial^2 F_1}{\partial z \partial v_\eta} F_2 + \frac{\partial F_1}{\partial z} \frac{\partial F_2}{\partial v_\eta} + \frac{\partial^2 F_1}{\partial v \partial v_\eta} v_\eta + \frac{\partial F_1}{\partial v} + \frac{\partial^2 F_1}{\partial v_\xi \partial v_\eta} \Omega(v, v_\xi^1, v_\eta^1) + \frac{\partial F_1}{\partial v_\xi} \frac{\partial \Omega(v, v_\xi^1, v_\eta^1)}{\partial v_\eta} = 0.$$

Taking into account equalities (7), (8), we have $\frac{\partial F_1}{\partial v_\eta} = 0$, $\frac{\partial F_2}{\partial v_\xi} = 0$ and then $\frac{\partial^2 F_1}{\partial z \partial v_\eta} = 0$, and equality remains

$$\frac{\partial F_1}{\partial z} \frac{\partial F_2}{\partial v_\eta} + \frac{\partial F_1}{\partial v} + \frac{\partial F_1}{\partial v_\xi} \frac{\partial \Omega(v, v_\xi^1, v_\eta^1)}{\partial v_\eta} = 0.$$

Having performed re-differentiation to $v_\eta$, we have

$$\frac{\partial^2 \Omega(v, v_\xi^1, v_\eta^1)}{\partial v_\eta^2} = 0,$$

$$\frac{\partial^2 F_1}{\partial z \partial v_\eta}\frac{\partial F_2}{\partial v_\eta} + \frac{\partial F_1}{\partial z}\frac{\partial^2 F_2}{\partial v_\eta^2} + \frac{\partial^2 F_1}{\partial v \partial v_\eta} + \frac{\partial^2 F_1}{\partial v_\xi \partial v_\eta}\frac{\partial \Omega(v, v_\xi^1, v_\eta^1)}{\partial v_\eta} = 0.$$

Considering $\frac{\partial F_1}{\partial v_\eta} = 0$, we obtain

$$\frac{\partial F_1}{\partial z}\frac{\partial^2 F_2}{\partial v_\eta^2} = 0.$$

We conduct similar actions with equality (3). Differentiating to $v_\xi$ twice, we obtain

$$\frac{\partial F_2}{\partial v}\frac{\partial^2 F_1}{\partial v_\xi^2} = 0.$$

Therefore, the functions $F_1$ and $F_2$ have a linear form to $v_\eta$ and $v_\xi$, respectively. Then, we have

$$\frac{\partial z}{\partial \xi} = f_1(z, v) + p_1(z, v)\frac{\partial v}{\partial \xi}, \tag{15}$$

$$\frac{\partial z}{\partial \eta} = f_2(z, v) + p_2(z, v)\frac{\partial v}{\partial \eta}. \tag{16}$$

We write the compatibility condition of Equation (6) with the new conditions (15) and (16):

$$\begin{aligned} (p_1 - p_2)\Omega(v, v_\xi^1, v_\eta^1) - \frac{\partial(f_2 + p_2 v_\eta)}{\partial v}v_\xi \\ + \frac{\partial(f_1 + p_1 v_\xi)}{\partial v}v_\eta + (f_2 + p_2 v_\eta)\frac{\partial(f_1 + p_1 v_\xi)}{\partial z} - (f_1 + p_1 v_\xi)\frac{\partial(f_2 + p_2 v_\eta)}{\partial z} = 0 \end{aligned} \tag{17}$$

After differentiating this expression to variable $v_\eta$ and $v_\xi$, we proceed to the analysis of the equation

$$(p_1 - p_2)\frac{\partial^2 \Omega(v, v_\xi^1, v_\eta^1)}{\partial v_\xi \partial v_\eta} - \frac{\partial p_2}{\partial v} + \frac{\partial p_1}{\partial v} + p_2\frac{\partial p_1}{\partial z} - p_1\frac{\partial p_2}{\partial z} = 0. \tag{18}$$

Further studies depend significantly on $\frac{\partial^2 \Omega(v, v_\xi^1, v_\eta^1)}{\partial v_\xi \partial v_\eta}$, so let us move on to a detailed analysis of equality (18) for each equation studied separately.

## 3. Results
### 3.1. Bäcklund Transformations for Nonlinear Equation

Let us perform the Bäcklund transformation for nonlinear Equation (10).

Equation (18) for (4), (5), considering that $\frac{\partial^2 \Omega(v, v_\xi^1, v_\eta^1)}{\partial v_\xi \partial v_\eta} = -1$ takes the form

$$\frac{\partial p_1}{\partial v} + p_2\frac{\partial p_1}{\partial z} - p_1 = p_1\frac{\partial p_2}{\partial z} + \frac{\partial p_2}{\partial v} - p_2. \tag{19}$$

It can be assumed that $p_1 \neq p_2$, and we define the relationship between the functions $p_1(z, v)$ and $p_2(z, v)$. We convert (19) to the following form:

$$\frac{\partial(p_1 - p_2)}{\partial v} - (p_1 - p_2) = p_1^2\frac{\partial}{\partial z}\frac{p_2}{p_1},$$

then, if we assume $p_1 - p_2 = e^v \varphi(z)$, then $p_1 = p_2 + e^v \varphi(z)$, and for the function $p_2$, we have the equation

$$[p_2 + e^v \varphi(z)]^2 \frac{\partial}{\partial z}\frac{p_2}{p_2 + e^v \varphi(z)} = 0.$$

Obviously, if $p_2 + e^v \varphi(z) = 0$, then $p_1 = 0$. This option could be considered but with only one undefined function $\varphi(z)$ remaining, which reduces the possibility of varying the unknowns in further reasoning, so calculate $p_2 + e^v \varphi(z) \neq 0$, and then

$$\frac{\partial}{\partial z} \frac{p_2}{p_2 + e^v \varphi(z)} = 0.$$

This leads to the dependence $\frac{p_2}{p_2 + e^v \varphi(z)} = \psi(v)$ and the definition of functions $p_2$ and $p_1$ in the form

$$p_2 = \frac{\psi(v)}{1 - \psi(v)} e^v \varphi(z), \quad p_1 = \frac{1}{1 - \psi(v)} e^v \varphi(z).$$

Now, Equation (17) will take the form

$$\left[ (a + bv)e^{2v}\varphi(z) - \frac{\partial f_2}{\partial v} + \frac{1}{1 - \psi(v)} e^v \left( f_2 \frac{\partial \varphi(z)}{\partial z} - \varphi(z) \frac{\partial f_2}{\partial z} \right) \right] v_\xi$$
$$+ f_2 \frac{\partial f_1}{\partial z} - f_1 \frac{\partial f_2}{\partial z} + \left[ \frac{\psi(v)}{1 - \psi(v)} e^v \left( \varphi(z) \frac{\partial f_1}{\partial z} - f_1 \frac{\partial \varphi(z)}{\partial z} \right) + \frac{\partial f_1}{\partial v} \right] v_\eta = 0.$$

We differentiate the last equation to the variable $v_\eta$ and the same expression to the variable $v_\xi$; as a result, we obtain the system

$$(a + bv)e^{2v}\varphi(z) - \frac{\partial f_2}{\partial v} + \frac{1}{1 - \psi(v)} e^v \left( f_2 \frac{\partial \varphi(z)}{\partial z} - \varphi(z) \frac{\partial f_2}{\partial z} \right) = 0, \tag{20}$$

$$f_2 \frac{\partial f_1}{\partial z} - f_1 \frac{\partial f_2}{\partial z} = 0, \tag{21}$$

$$\frac{\psi(v)}{1 - \psi(v)} e^v \left( \varphi(z) \frac{\partial f_1}{\partial z} - f_1 \frac{\partial \varphi(z)}{\partial z} \right) + \frac{\partial f_1}{\partial v} = 0. \tag{22}$$

We look for functions $f_1(z, v)$, $f_2(z, v)$ in the following form

$$f_1(z, v) = \psi_1(v) g_1(z), \quad f_2(z, v) = \psi_2(v) g_2(z).$$

We substitute these equations in the system (20)–(22) and isolate the logarithmic derivatives $\ln g_2$, $\ln g_1$, $\ln \varphi$:

$$(a + bv)e^{2v}\varphi(z) - g_2(z) \frac{\partial \psi_2(v)}{\partial v} + \frac{\psi_2(v)}{1 - \psi(v)} e^v g_2(z) \varphi(z) \frac{\partial}{\partial z} \left( \ln \frac{\varphi(z)}{g_2(z)} \right) = 0, \tag{23}$$

$$\psi_2(v)\psi_1(v)g_2(z)g_1(z) \frac{\partial}{\partial z} \left( \ln \frac{g_1(z)}{g_2(z)} \right) = 0, \tag{24}$$

$$g_1(z) \frac{\partial \psi_1}{\partial v} - \frac{\psi(v)\psi_1(v)}{1 - \psi(v)} e^v \varphi(z) g_1(z) \frac{\partial}{\partial z} \left( \ln \frac{\varphi(z)}{g_1(z)} \right) = 0. \tag{25}$$

One can choose a special form of functions $g_j(z)$, $\varphi(z)$ so that the system takes a simpler form; then, the differential equations can be explicitly integrated. We calculate

$$g_1(z) = g_2(z) = k_1 z, \quad \varphi(z) = k_2 z, \quad k_1, k_2 = \text{const},$$

and the system (23)–(25) will take the form

$$k_2(a + bv)e^{2v}z - k_1 \frac{\partial \psi_2(v)}{\partial v} = 0,$$

$$k_1 z \frac{\partial \psi_1}{\partial v} = 0.$$

As a result, simple differential equations are obtained for functions $\psi_2(v)$, $\psi_1(v)$. Let us define them:

$$\psi_2 = \frac{k_2}{k_1}\int (a+bv)e^{2v}dv = \frac{k_2}{4k_1}(2a-b+2bv)e^{2v}+C_1, \quad \psi_1 = k = \text{const.}$$

Now, the transformations (2), (3) take the form

$$\frac{\partial z}{\partial \xi} = kk_1 z + \frac{k_2}{1-\psi(v)}e^v z \frac{\partial v}{\partial \xi}, \tag{26}$$

$$\frac{\partial z}{\partial \eta} = z\left[\frac{k_2}{4}(2a-b+2bv)e^{2v}+C_1\right] + k_2 e^v z \frac{\psi(v)}{1-\psi(v)}\frac{\partial v}{\partial \eta}. \tag{27}$$

Thus, the Bäcklund transformation is obtained in the form (26), (27). The system (26), (27) is combined with any function $\psi(v)$. We consider the following option: $\psi(v) = 2$, $C_1 = 0$, $k = k_1 = 1$, $k_2 = -2$, and then the relations (26), (27) will take the form

$$\frac{\partial z}{\partial \xi} = z + 2e^v z \frac{\partial v}{\partial \xi}, \quad \frac{\partial z}{\partial \eta} = 4e^v z \frac{\partial v}{\partial \eta} - \left(a - \frac{1}{2}b + bv\right)ze^{2v}. \tag{28}$$

Let us check whether it is possible to obtain Equation (10) from the system (28).

If we differentiate the first equality of the system (28) to the variable $\eta$ and the second to the variable $\xi$, we obtain

$$z_{\xi\eta} = z_\eta + 2e^v z v_\eta v_\xi + 2e^v z_\eta v_\xi + 2e^v z v_{\eta\xi},$$

$$z_{\xi\eta} = 4e^v z v_\xi v_\eta + 4e^v z_\xi v_\eta + 4e^v z v_{\xi\eta} - 2(a+bv)ze^{2v}v_\xi - \left(a - \frac{1}{2}b + bv\right)z_\xi e^{2v}.$$

We subtract the upper equality from the lower one and collect similar terms:

$$(1+2e^v v_\xi)z_\eta = 2e^v z v_{\xi\eta} + 2e^v z v_\xi v_\eta - 2(a+bv)ze^{2v}v_\xi + \left[4v_\eta - \left(a - \frac{1}{2}b + bv\right)e^v\right]e^v z_\xi.$$

We eliminate the derivatives $z_\eta$, $z_\xi$, using the relations (28). Canceling by non-zero functions, we obtain Equation (10).

Let us see which equation goes to the original Equation (10) using transformations (28). To do this, we convert Equation (28) to the form

$$\frac{\partial \ln z}{\partial \xi} = 1 + 2\frac{\partial e^v}{\partial \xi}, \quad \frac{\partial \ln z}{\partial \eta} = 2\frac{\partial e^v}{\partial \eta} - \left(a - \frac{1}{2}b + bv\right)e^{2v}. \tag{29}$$

Let us try to identify how the functions $z(\xi,\eta)$ and $v(\xi,\eta)$ are related, taking into account that the functions satisfy Equation (10). Let us differentiate the second equality (29) to the variable $\xi$:

$$(\ln z)_{\eta\xi} = 4(e^v)_{\eta\xi} - 2(a+bv)e^{2v}v_\xi$$

and replace the expression $(a+bv)e^v v_\xi$ with the terms of Equation (10), then

$$(\ln z)_{\eta\xi} = 4(e^v)_{\eta\xi} - 2(v_{\eta\xi} + v_\xi v_\eta)e^v = 2(e^v)_{\eta\xi}.$$

Taking into account the first differential constraint (29), the derivatives can be omitted up to a constant

$$\ln z = 2e^v + \xi.$$

Then, the function $v(\xi,\eta)$ is expressed through $z(\xi,\eta)$

$$v = \ln\left[\frac{1}{2}(\ln z - \xi)\right]. \tag{30}$$

Denoting $\ln z = w(\xi, \eta)$ and substituting (30) in (10), we obtain

$$w_{\eta\xi} = \frac{1}{2}\left(a + b\ln\left[\frac{1}{2}(w - \xi)\right]\right)(w - \xi)(w_\xi - 1). \tag{31}$$

**Theorem 1.** *Bäcklund transformations*

$$\frac{\partial w}{\partial \eta} = 2e^v\frac{\partial v}{\partial \eta} - \left(a - \frac{1}{2}b + bv\right)e^{2v}, \ \frac{\partial w}{\partial \xi} = 1 + 2e^v\frac{\partial v}{\partial \xi}, \tag{32}$$

*link Equations (10)–(31).*

Equation (11) is a special case of (10), so the following conclusion can be formulated for this equation.

**Corollary 1.** *Bäcklund transformations*

$$\frac{\partial w}{\partial \xi} = 1 + 2e^v\frac{\partial v}{\partial \xi}, \ \frac{\partial w}{\partial \eta} = 4e^v\frac{\partial v}{\partial \eta} - \frac{\alpha_{32}\alpha_{21}}{4\alpha_{11}}ve^{2v}, \tag{33}$$

*link Equation (11) to the following equation:*

$$\frac{\alpha_{32}\alpha_{21}}{8\alpha_{11}}(w_\xi - 1)(w - \xi)\left(\frac{1}{2} + \ln\left[\frac{1}{2}(w - \xi)\right]\right) - w_{\xi\eta} = 0. \tag{34}$$

**Theorem 2.** *For Equation (11), there is a Bäcklund auto-transformation of the form*

$$e^g\frac{\partial g}{\partial \xi} = e^v\frac{\partial v}{\partial \xi}, \ e^g\frac{\partial g}{\partial \eta} = 2e^v\frac{\partial v}{\partial \eta} - \frac{\alpha_{32}\alpha_{21}}{8\alpha_{11}}ve^{2v}. \tag{35}$$

**Proof of Theorem 2.** Let us write the equality (35) in the following form:

$$\frac{\partial e^g}{\partial \xi} = e^v\frac{\partial v}{\partial \xi}, \qquad \frac{\partial e^g}{\partial \eta} = 2e^v\frac{\partial v}{\partial \eta} - \frac{\alpha_{32}\alpha_{21}}{8\alpha_{11}}ve^{2v},$$

and cross-differentiate. Equalizing the left parts gives

$$e^v\frac{\partial v}{\partial \xi}\frac{\partial v}{\partial \eta} + e^v\frac{\partial^2 v}{\partial \xi \partial \eta} - \frac{\alpha_{32}\alpha_{21}}{8\alpha_{11}}\frac{\partial v}{\partial \xi}e^{2v} - \frac{\alpha_{32}\alpha_{21}}{4\alpha_{11}}\frac{\partial v}{\partial \xi}ve^{2v} = 0$$

or Equation (11). $\square$

Now, we rewrite the second equality (35) in the form

$$\frac{\partial e^g}{\partial \eta} = 2\frac{\partial e^v}{\partial \eta} - \frac{\alpha_{32}\alpha_{21}}{8\alpha_{11}}ve^{2v},$$

and differentiate by $\xi$

$$\frac{\partial^2 e^g}{\partial \eta \partial \xi} = 2\frac{\partial^2 e^v}{\partial \eta \partial \xi} - \frac{\alpha_{32}\alpha_{21}}{8\alpha_{11}}e^{2v}(1 - 2v)\frac{\partial v}{\partial \xi}.$$

We replace the term $\frac{\alpha_{32}\alpha_{21}}{4\alpha_{11}}v_\xi e^v(1 + 2v)$ in the last equality with the remaining terms from (11), and then

$$\frac{\partial^2 e^g}{\partial \eta \partial \xi} = 2\frac{\partial^2 e^v}{\partial \eta \partial \xi} - e^v\left(\frac{\partial^2 v}{\partial \eta \partial \xi} + \frac{\partial v}{\partial \xi}\frac{\partial v}{\partial \eta}\right),$$

which leads to the equality

$$\frac{\partial^2 e^g}{\partial \eta \partial \xi} = \frac{\partial^2 e^v}{\partial \eta \partial \xi}.$$

This means that the functions $e^g$ and $e^v$ can differ only by arbitrary terms of the form $\varphi(\xi) + \psi(\eta)$, so

$$e^g + \varphi(\xi) + \psi(\eta) = e^v. \tag{36}$$

If $\varphi(\xi) = \psi(\eta) = 0$, then $g = v$, and from equalities (35), we obtain Equation (11).

Let us determine what happens if $\varphi(\xi) \neq 0$, $\psi(\eta) \neq 0$. Let us perform substitution (33) in Equation (11); then, we obtain

$$e^g_{\eta\xi} = \frac{\alpha_{32}\alpha_{21}}{8\alpha_{11}}([e^g + \varphi(\xi) + \psi(\eta)]^2 \ln[e^g + \varphi(\xi) + \psi(\eta)])_\xi.$$

Let us perform differentiation

$$e^g(g_{\eta\xi} + g_\xi g_\eta) = \frac{\alpha_{32}\alpha_{21}}{8\alpha_{11}} 2[e^g + \varphi(\xi) + \psi(\eta)] \ln[e^g + \varphi(\xi) + \psi(\eta)](e^g g_\xi + \varphi'(\xi)) \\ + \frac{\alpha_{32}\alpha_{21}}{8\alpha_{11}}[e^g + \varphi(\xi) + \psi(\eta)](e^g g_\xi + \varphi'(\xi))$$

and group the terms with a common derivative; we obtain

$$e^g(g_{\eta\xi} + g_\xi g_\eta) = \frac{\alpha_{32}\alpha_{21}}{8\alpha_{11}}[e^g + \varphi(\xi) + \psi(\eta)](2\ln[e^g + \varphi(\xi) + \psi(\eta)] + 1)(e^g g_\xi + \varphi'(\xi)),$$

here, $\varphi(\xi)$, $\psi(\eta)$ are arbitrary functions.

**Corollary 2.** *Bäcklund transformations*

$$\frac{\partial q}{\partial \xi} + \varphi'(\xi) = e^v \frac{\partial v}{\partial \xi},$$

$$\frac{\partial q}{\partial \eta} + \psi'(\eta) = 2e^v \frac{\partial v}{\partial \eta} - \frac{\alpha_{32}\alpha_{21}}{8\alpha_{11}} v e^{2v},$$

*associate Equation (11) with the equation*

$$q_{\eta\xi} = \frac{\alpha_{32}\alpha_{21}}{8\alpha_{11}}[q + \varphi(\xi) + \psi(\eta)](2\ln[q + \varphi(\xi) + \psi(\eta)] + 1)(q_\xi + \varphi'(\xi)),$$

*here, $\varphi(\xi)$, $\psi(\eta)$ are arbitrary functions.*

Similarly, starting the transformation with a detailed analysis of equality (18), in each case for the remaining studied Equations (12) and (13), the following theorems are proved:

**Theorem 3.** *Bäcklund transformations of the form*

$$w_{\xi\xi} = w_\xi \frac{\partial v}{\partial \xi} - \frac{\alpha_{21}}{16\alpha_{11}^2} e^v w_\xi, \tag{37}$$

$$w_{\xi\eta} = \frac{w_\xi}{2} \frac{\partial v}{\partial \eta} - \frac{\alpha_{21}}{16\alpha_{11}^2} e^v w_\xi, \tag{38}$$

*connect Equation (12) with the equation*

$$(w^2)_{\xi\eta} = 4w_\xi^2. \tag{39}$$

**Theorem 4.** *Bäcklund transformations of the form*

$$\frac{\partial w}{\partial \xi} = \frac{\partial v}{\partial \xi} - e^v, \ \frac{\partial w}{\partial \eta} = e^{-v}$$

*connect Equation (13) with the equation*

$$w_{\eta\xi} + w_\xi w_\eta = -1.$$

*3.2. Applying Differential Couplings to Obtain Exact Solutions*

**Theorem 5.** *If Equation (39) has a solution*

$$w = 2\eta + \xi, \tag{40}$$

*then Equation (12) has a solution:*

$$v = -\ln\left[C - \frac{\alpha_{21}}{16\alpha_{11}^2}(\xi + \eta)\right], \quad C = \text{const.} \tag{41}$$

**Proof of Theorem 5.** We use the found transformations (37), (38) and substitute the known solution (40) in them, and then system (37), (38) takes the form

$$\frac{\partial v}{\partial \xi} = \frac{\alpha_{21}}{16\alpha_{11}^2}e^v, \qquad \frac{\partial v}{\partial \eta} = \frac{\alpha_{21}}{16\alpha_{11}^2}e^v,$$

from here, we find

$$e^{-v} = C - \frac{\alpha_{21}}{16\alpha_{11}^2}(\xi + \eta),$$

here, *C* is an arbitrary constant. As a result, the solution (41) of Equation (12) was found. □

Let us perform some transformations in Equation (39), multiplying both sides by $w^2$:

$$w^2(w^2)_{\xi\eta} - \left[(w^2)_\xi\right]^2 = 0,$$

and, using the Fourier method of separation of variables, we obtain a solution to Equation (39) in the form

$$w = e^{\frac{\lambda}{2}(\eta+\xi)}, \quad \lambda = \text{const.}$$

**Theorem 6.** *If Equation (39) has a solution*

$$w = e^{\frac{\lambda}{2}(\eta+\xi)}, \quad \lambda = \text{const} \tag{42}$$

*then Equation (12) has a solution*

$$v = \ln\lambda + \lambda\left(\eta + \frac{1}{2}\xi\right) - \ln\left|1 - \frac{\alpha_{21}}{8\alpha_{11}^2}e^{\lambda[\eta+\frac{1}{2}\xi]}\right|. \tag{43}$$

**Proof of Theorem 6.** Using the found transformations (37), (38), we substitute the known solution (42) into it, and then system (37), (38), after cancellation by $\frac{\lambda}{2}e^{\frac{\lambda}{2}(\eta+\xi)}$, takes the form

$$\frac{\lambda}{2} = \frac{\partial v}{\partial \xi} - \frac{\alpha_{21}}{16\alpha_{11}^2}e^v, \qquad \lambda = \frac{\partial v}{\partial \eta} - \frac{\alpha_{21}}{8\alpha_{11}^2}e^v, \tag{44}$$

from the first linear partial differential equation, we find

$$v - \ln\left[\frac{\lambda}{2} + \frac{\alpha_{21}}{16\alpha_{11}^2}e^v\right] = \frac{\lambda}{2}\xi + \varphi(\eta),$$

where $\varphi(\eta)$ is an arbitrary function, and from the second equation of system (44), we determine the form of the function $\varphi(\eta)$: $\varphi(\eta) = 2e^{\lambda\eta}$. $\square$

Expressing the function $v$ explicitly, we obtain the solution (43) of Equation (12).

**Theorem 7.** *If Equation (12) has a solution*

$$v = a(\eta + \xi),$$

*then Equation (39) has a solution*

$$w = -\frac{16\alpha_{11}^2}{\alpha_{21}}\exp\left[-\frac{\alpha_{21}}{16\alpha\alpha_{11}^2}e^{\alpha(\eta+\xi)} - \frac{\alpha}{2}\eta\right]. \tag{45}$$

**Proof of Theorem 7.** Using the found transformations (37), (38), we substitute the known solution $v = a(\eta + \xi)$, and then we obtain the system of equations

$$(\ln w_\xi)_\xi = \alpha - \frac{\alpha_{21}}{16\alpha_{11}^2}e^{\alpha(\eta+\xi)},$$

$$(\ln w_\xi)_\eta = \frac{\alpha}{2} - \frac{\alpha_{21}}{16\alpha_{11}^2}e^{\alpha(\eta+\xi)},$$

which can be easily integrated over the corresponding variables

$$\ln w_\xi = \alpha\xi - \frac{\alpha_{21}}{16\alpha\alpha_{11}^2}e^{\alpha(\eta+\xi)} + \varphi(\eta), \tag{46}$$

$$\ln w_\xi = \frac{\alpha}{2}\eta - \frac{\alpha_{21}}{16\alpha\alpha_{11}^2}e^{\alpha(\eta+\xi)} + \psi(\xi), \tag{47}$$

where $\varphi(\eta), \psi(\xi)$ are the constants of integration (arbitrary functions). $\square$

Let us extend the definition of the functions $\varphi(\eta)$ and $\psi(\xi)$ so that the obtained values of the right-hand sides of the system (46), (47) coincide as follows:

$$\varphi(\eta) = \frac{\alpha}{2}\eta, \quad \psi(\xi) = \alpha\xi.$$

As a result, an expression for the function $w_\xi$ is defined:

$$w_\xi = e^{\alpha(\xi+\frac{1}{2}\eta) - \frac{\alpha_{21}}{16\alpha\alpha_{11}^2}e^{\alpha(\eta+\xi)}}.$$

We perform integration over $\xi$, and we obtain the unknown function $w(\xi, \eta)$ (45). The form of function (45) is shown from two angles in Figure 1 (for $\alpha = 1$, $\frac{\alpha_{21}}{\alpha_{11}^2} = -32$).

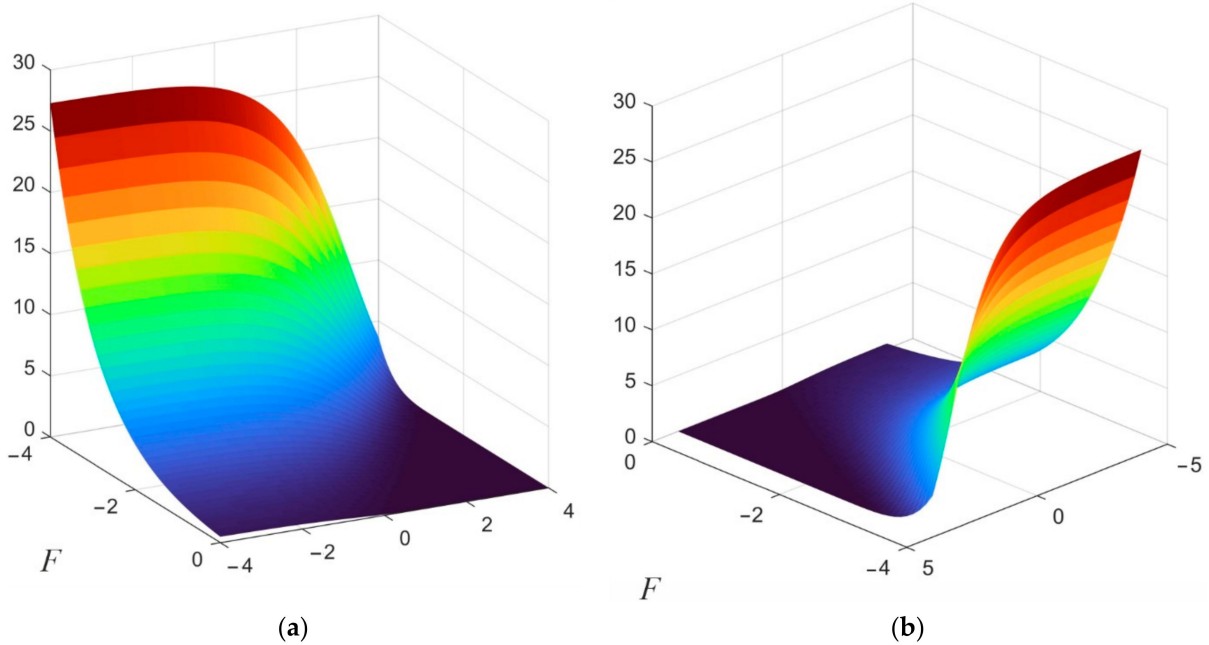

**Figure 1.** A 3D graph of the function (45) shown from two angles (**a**) and (**b**). Here, $F \equiv w(\xi, \eta)$.

**Theorem 8.** *Equation (31) has a solution implicitly given in the form of a series*

$$\ln \left| \ln \left[ \frac{1}{2}(w - \xi) \right] \right| + \sum_{n=1}^{\infty} \frac{(-1)^n}{n \cdot n!} \ln^n \left[ \frac{1}{2}(w - \xi) \right] = \frac{\alpha_{32} \alpha_{21}}{8 \alpha_{11}} (\gamma \xi + \eta) + C_2,$$

*where constants $\gamma$, $C_2$ are arbitrary constants.*

The proof is carried out by simple verification.

**Theorem 9.** *If (10) has a solution $v = a$, then (31) has a solution*

$$w = \xi - \frac{\alpha_{32} \alpha_{21}}{4 \alpha_{11}} a e^{2a} \eta. \tag{48}$$

**Proof of Theorem 9.** We substitute $v = a$ into the found differential links (34) and integrate each equality.

$$w = \xi + \varphi(\eta),$$

$$w = -\frac{\alpha_{32} \alpha_{21}}{4 \alpha_{11}} a e^{2a} \eta + \psi(\xi).$$

We equate the obtained expressions for the function $w$ and redefine arbitrary functions $\varphi(\eta), \psi(\xi)$. As a result, we obtain (48). □

**Theorem 10.** *If Equation (11) has a solution $v = \eta$, then Equation (34) has a solution*

$$w = 4e^{\eta} - \frac{\alpha_{32} \alpha_{21}}{16 \alpha_{11}} (2\eta - 1) e^{2\eta} + \xi. \tag{49}$$

**Proof of Theorem 10.** Substitute $v = \eta$ into the found differential constraints (33) and integrate each equality.

$$w = \xi + \varphi(\eta),$$

$$w = 4e^\eta - \frac{\alpha_{32}\alpha_{21}}{16\alpha_{11}}(2\eta - 1)e^{2\eta} + \psi(\xi).$$

We equate the obtained expressions for the function $w$ and redefine arbitrary functions $\varphi(\eta), \psi(\xi)$. As a result, we obtain (49) (Figure 2). $\square$

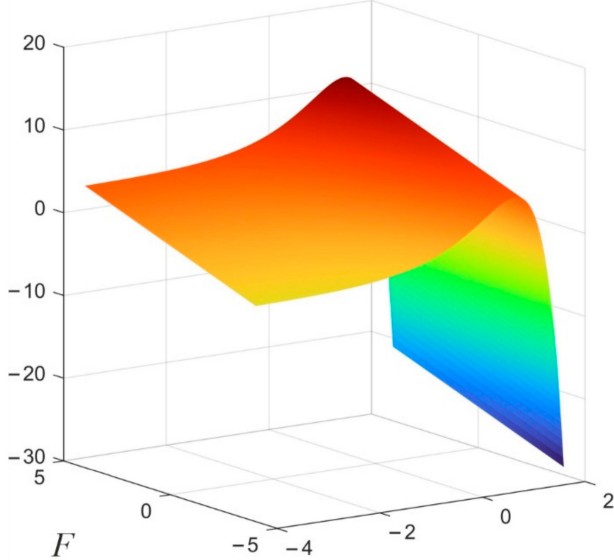

**Figure 2.** The plot is according to Formula (49) at $\frac{\alpha_{32}\alpha_{21}}{16\alpha_{11}} = \frac{1}{3}$. Here, $F \equiv w(\xi, \eta)$.

**Theorem 11.** *Equation (11) has a solution implicitly given in the form of a series*

$$\ln|v| + \sum_{n=1}^{\infty} \frac{(-v)^n}{n \cdot n!} = \frac{\alpha_{32}\alpha_{21}}{8\alpha_{11}}(\gamma\xi + \eta) + C_2, \tag{50}$$

*where constants $\gamma, C_2$ are arbitrary constants.*

The proof is carried out by simple verification.

Solution (50) is a cylindrical surface with a guide shown in Figure 3.

Let us use the found auto-Bäcklund transformations (35) for Equation (11) and solution (50). If we assume that $g(\xi, \eta) = v(\xi, \eta)$, then using (35), we can find a new solution to Equation (11). Substitute expression (50) into the left-hand side of (35):

$$\frac{\alpha_{32}\alpha_{21}}{8\alpha_{11}}g = \frac{1}{\gamma}e^v\frac{\partial v}{\partial \xi},$$

$$\frac{\alpha_{32}\alpha_{21}}{8\alpha_{11}}g = 2e^v\frac{\partial v}{\partial \eta} - \frac{\alpha_{32}\alpha_{21}}{8\alpha_{11}}ve^{2v}.$$

Equating the left-hand sides, we obtain a linear first-order equation for the function $v(\xi, \eta)$

$$2\frac{\partial v}{\partial \eta} - \frac{1}{\gamma}\frac{\partial v}{\partial \xi} = \frac{\alpha_{32}\alpha_{21}}{8\alpha_{11}}ve^v, \tag{51}$$

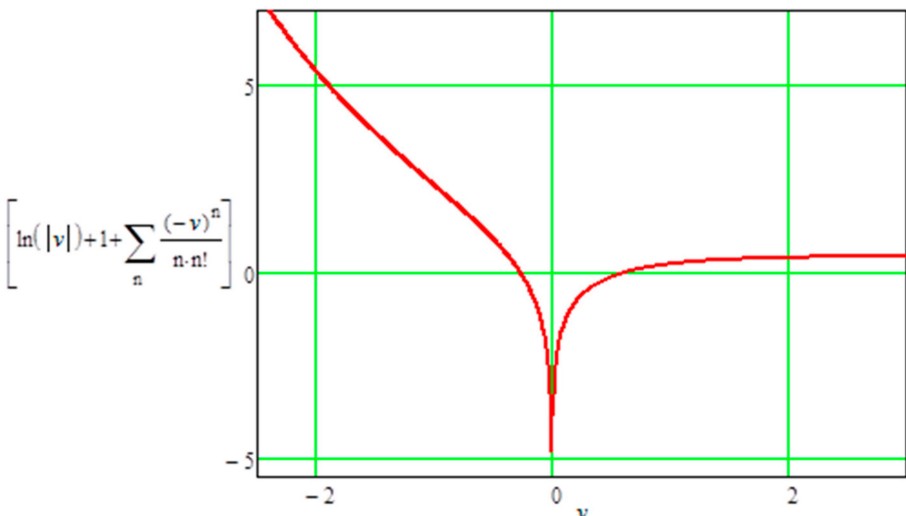

**Figure 3.** Cylindrical surface guide (50), where $n = 1, 2, \ldots, 100$.

To find the general solution of this equation, we find the first integrals of the system

$$2\gamma\xi + \eta = C_1, \qquad \frac{8\alpha_{11}}{\gamma\alpha_{32}\alpha_{21}}\left[\ln|v| + \sum_{n=1}^{\infty}\frac{(-v)^n}{n\cdot n!}\right] + \xi = C_2. \tag{52}$$

The general solution to equation (51) has the form

$$F\left(2\gamma\xi + \eta, \ \frac{8\alpha_{11}}{\gamma\alpha_{32}\alpha_{21}}\left[\ln|v| + \sum_{n=1}^{\infty}\frac{(-v)^n}{n\cdot n!}\right] + \xi\right) = C,$$

Here, $F$ is any function.

Equation (11) is nonlinear; therefore, it is necessary to clarify the form of the function $F$. Let us substitute expression (52) into Equation (11)

$$\frac{16\alpha_{11}}{\alpha_{32}\alpha_{21}}\left[2\frac{F_1}{F_2^2}F_{12} - \frac{F_1^2}{F_2^3}F_{22} - \frac{F_{11}}{F_2}\right] = -\left(\frac{1}{\gamma} + \frac{F_1}{F_2}\right)\left(2\gamma\frac{F_1}{F_2} + 1\right)e^v(1 + 2v). \tag{53}$$

Here, $F_1$ is the derivative of $F$ by the first component, and $F_2$ is the derivative of $F$ by the second component.

As one can see, equality (53) is not identically fulfilled; therefore, it is necessary to require that one of the systems is fulfilled:

$$\begin{cases} \frac{1}{\gamma} + \frac{F_1}{F_2} = 0, \\ 2\frac{F_1}{F_2^2}F_{12} - \frac{F_1^2}{F_2^3}F_{22} - \frac{F_{11}}{F_2} = 0, \end{cases} \text{, or} \begin{cases} \frac{F_1}{F_2} = -\frac{1}{2\gamma}, \\ 2\frac{F_1}{F_2^2}F_{12} - \frac{F_1^2}{F_2^3}F_{22} - \frac{F_{11}}{F_2} = 0 \end{cases}. \tag{54}$$

All the terms of the equalities are homogeneous, so we use the technique that allows us to separate the arguments of the function. We represent $F$ in the form

$$F = X(C_1)Y(C_2),$$

where $X$ depends on the first component $C_1$ of the function $F$, and $Y$ on the second component $C_2$ (52), and then the first equality of system (54) takes the following form (due to the similarity of the first equalities of the systems, the result of the substitution for the second system is written in parentheses):

$$\frac{X'}{X} = -\frac{1}{\gamma}\frac{Y'}{Y} = \lambda, \quad \left(\frac{X'}{X} = -\frac{1}{2\gamma}\frac{Y'}{Y} = \lambda\right),$$

Here, $\lambda$ is an arbitrary parameter.

The functions take the form

$$\ln|X| = \lambda C_1, \quad \ln|Y| = -\gamma\lambda C_2, \quad (\ln|X| = \lambda C_1, \quad \ln|Y| = -2\gamma\lambda C_2),$$

which leads to the following kind of function

$$F(C_1, C_2) = e^{\lambda C_1 - \gamma\lambda C_2}, \quad \left(F(C_1, C_2) = e^{\lambda C_1 - 2\gamma\lambda C_2}\right), \tag{55}$$

where

$$\lambda(C_1 - \gamma C_2) = \lambda\left(\gamma\xi + \eta - \frac{8\alpha_{11}}{\alpha_{32}\alpha_{21}}\left[\ln|v| + \sum_{n=1}^{\infty}\frac{(-v)^n}{n \cdot n!}\right]\right),$$

$$\left(\lambda(C_1 - 2\gamma C_2) = \lambda\eta - \lambda\frac{8\alpha_{11}}{\alpha_{32}\alpha_{21}}\left[\ln|v| + \sum_{n=1}^{\infty}\frac{(-v)^n}{n \cdot n!}\right]\right).$$

The connection between the components $C_1, C_2$, satisfying the second system (expression in brackets), led to the absence of dependence on the variable $\xi$, so this case is not considered further.

The second equality of system (54) is satisfied identically. The dependence on $\lambda$ is insignificant; therefore, we assume $\lambda = 1$.

The following theorem is proved:

**Theorem 12.** *Equation (11) has a solution*

$$\exp\left(\gamma\xi + \eta - \frac{8\alpha_{11}}{\alpha_{32}\alpha_{21}}\left[\ln|v| + \sum_{n=1}^{\infty}\frac{(-v)^n}{n \cdot n!}\right]\right) = C.$$

The result of the theorem can be generalized.

**Corollary 3.** *Equation (11) has a solution*

$$F\left(\gamma\xi + \eta - \frac{8\alpha_{11}}{\alpha_{32}\alpha_{21}}\left[\ln|v| + \sum_{n=1}^{\infty}\frac{(-v)^n}{n \cdot n!}\right]\right) = C, \tag{56}$$

*here, F is an arbitrary function.*

The proof is carried out by simple verification.

## 4. Discussion

The considered equations refer to wave equations with a nonlinear right-hand side, which has an exponential–power relationship. The exponential–power model is a multiplicative combination of exponential and power models. Finding exact solutions to such equations is fraught with great difficulties since a change in variables does not bring the equation to a linear form or simplification; therefore, it is necessary to use a modification that differs from the mappings. Differential links are such a transformation. Bäcklund transformations are a differential relationship of two equations. Recently, this approach has made it possible to solve many interesting problems [8–11,14,17–19].

In addition, for a given solution of one equation, Bäcklund transformations make it possible to determine, up to a finite number of constants, the solution of the second equation, and this connection works in two directions. Therefore, for Equations (12) and (39), choosing a simple solution in the form $w = 2\eta + \xi$, and for Equation (39), using the constructed Bäcklund transformations (37), (38), a solution of Equation (12) was found in the form (41) (application of differential constraints (Statements 1 and 2)). Using the same differ-

ential constraints (37), (38) from the solution of Equation (12), an exact solution to Equation (39) was obtained (application of differential constraints (Statement 3)). Similar results were obtained for pairs of equations: Equations (10) and (31) and Equations (11) and (34).

An especially interesting case is when the Bäcklund transformations translate the equation into itself—auto-transformations. This property is typical for nonlinear equations with soliton solutions [13]. The present article discusses the construction of auto-transformations for Equation (11) (Section 3 (Results), Theorem 2). Differential constraints (35) made it possible to construct a general solution (56) from solution (50).

## 5. Conclusions

For the equations studied in the article, new equations were found using Bäcklund transformations, which make it possible to find solutions to the original nonlinear equations and to identify internal connections between various integrable equations.

The present paper proves theorems on Bäcklund transformations of nonlinear hyperbolic partial differential equations of the second order of the Klein–Gordon class, which are special cases of the Liouville equation, with exponential nonlinearity having a multiplier depending on the function and its first derivatives. The transformations were constructed using Clairin's method. The new equations obtained with the help of differential connections can be used for further studies of equations of this type, as well as for solving many applied problems in various fields of natural science.

**Author Contributions:** Conceptualization, methodology, investigation and writing—original draft preparation, T.V.R.; investigation and writing—original draft preparation, R.G.Z.; validation and writing—review and editing, A.R.Z.; validation and formal analysis, O.V.N. All authors have read and agreed to the published version of the manuscript.

**Funding:** This research was supported by the North-Caucasus Center for Mathematical Research under agreement No. 075-02-2021-1749 with the Ministry of Science and Higher Education of the Russian Federation.

**Conflicts of Interest:** The authors declare no conflict of interest.

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
