# Peer review of "Bäcklund Transformations for Liouville Equations with Exponential Nonlinearity"

_axioms, doi:10.3390/axioms10040337_

Round 1

Reviewer 1 Report

The authors study Bäcklund transformations to find solutions to nonlinear differential equations, in particular, for non-linear partial second-order derivatives of the soliton type with logarithmic nonlinearity and hyperbolic linear parts. The results and examples presented in the manuscript deserve to be published but to be accepted the manuscript needs some revision.                     On page 3 the authors write “We consider the function z to be known” and then on page 4 above Eq. (11) they write “…move to the simplest hyperbolic equation ... “. If a function z  is already known what is the meaning of the differential equation for it?

The language of the manuscript has to be improved significantly! There are also misprints, e.g. the equality sing instead of the minus sign after Eq. (15) on line 105.

To summarize, the manuscript can be accepted for publication after significant revision.

Author Response

Response to Reviewer 1 Comments

The authors are grateful to the referee for the comments made and are confident that the corrections will serve to better understand our results by the readers of the journal.

  1. The authors agree with the comment related to function z. The thought is not expressed correctly. We meant that the function is a solution to some simple equation, the form of which will be defined below. Corrections have been made to the text of the article.
  2. The authors agree with the remark related to the typos. The typos have been removed.
  3. The language of the manuscript has been edited.

Sincerely,

                                    Authors.

Author Response

Response to Reviewer 2 Comments

The authors are grateful to the referee for the comments made and are confident that the corrections will serve to better understand our results by the readers of the journal.

  1. The introduction has been rewritten in order to emphasize the novelty of the results obtained in comparison with other similar works.
  2. The abstract is corrected.
  3. Grammatical errors have been corrected.
  4. Figure captions have been corrected.
  5. The Discussion section has been expanded.

Sincerely,

                                    Authors.

Round 2

Reviewer 2 Report

I'm satisfied with the current version of the paper, except for one thing:
In Eq. (6), there is z tilde in the second term on the right-hand side, denominator. I've asked to correct or explain it, but the pdf maker tricked us and the formula is missing from my review report. So please fix it.